# The Endogenous Expression of BMI1 in Adult Human Eyes

**DOI:** 10.3390/cells13191672

**Published:** 2024-10-09

**Authors:** Zhongyang Lu, Maria G. Morales, Shufeng Liu, Hema L. Ramkumar

**Affiliations:** Oculogenex Inc., 2250 W Whittier Blvd., Suite 300, La Habra, CA 90631, USA; gmorales@oculogenex.com (M.G.M.); sliu@oculogenex.com (S.L.)

**Keywords:** BMI1, protein expression and localization, immunoassay, AMD

## Abstract

BMI1, also known as B lymphoma Mo-MLV insertion region 1, is a protein in the Polycomb group that is implicated in various cellular processes, including stem cell self-renewal and the regulation of cellular senescence. BMI1 plays a role in the regulation of retinal progenitor cells and the renewal of adult neuronal cells. However, the presence, location, and quantification of BMI1 in the adult human eye have never previously been reported. In this study, we collected 45 frozen globes from eye banks, and ocular tissues were dissected. Protein was quantified by utilizing a custom electrochemiluminescence (ECL) assay developed to quantify the BMI1 protein. BMI1 was found in all ocular tissues at the following levels: the retina (1483.6 ± 191.7 pg/mL) and the RPE (296.4 ± 78.1 pg/mL). BMI1 expression was noted ubiquitously in the GCL (ganglion cell layer), the INL (inner nuclear layer), the ONL (outer nuclear layer), and the RPE (retinal pigment epithelium) via immunofluorescence, with higher levels in the inner than in the outer retinal layers and the RPE. These data confirm that BMI1 is expressed in the human retina. Further studies will illuminate the role that BMI1 plays in ocular cells. BMI1 levels are lower in aged retinas, possibly reflecting changes in retinal somatic and stem cell maintenance and disease susceptibility.

## 1. Introduction

The human retina is a complex and delicate structure essential for vision, comprising various cell types that work in concert to process visual information. Maintaining the health and functionality of the retina is crucial, as its degeneration can lead to severe visual impairment or blindness. BMI1 (B lymphoma Mo-MLV insertion region 1) is a polycomb group protein that plays a significant role in the regulation of neural stem cell self-renewal, cellular homeostasis, the regulation of aging and senescence, and protection against oxidative stress in multiple cell types [1,2,3,4]. In the retina, BMI1 has emerged as a critical factor in maintaining retinal homeostasis, promoting regenerative processes, and protecting against cellular damage [5,6].

Research has shown that BMI1 is involved in the maintenance of retinal progenitor cells and retinal stem cells, playing a role in retinal repair and regeneration [1,7,8]. BMI1 regulates mitochondrial oxidative stress by upregulating multiple antioxidants and downregulating *p*53, regulating the cell’s reaction to oxidative stress. Its functions in regulating the balance between cell repair and senescence are vital for responding to, and repairing, cumulative oxidative stress damage. Furthermore, BMI1’s role in managing oxidative stress helps shield retinal cells from the detrimental effects of reactive oxygen species, which are abundant in the metabolically active environment of the retina.

BMI1 knockout mice have a premature aging phenotype with cataracts [5], senescent retinal cells, and cone photoreceptor degeneration [9,10]. These mice have many features of the dry AMD phenotype. Alterations in BMI1 expression or function are linked to various degenerative diseases such as Alzheimer’s disease [11] and premature ovarian failure [12,13,14]. Downregulation or dysfunction of BMI1 can impair the regenerative capacity of the retina and lead to cone necroptosis, a hallmark of degenerative retinal diseases such as age-related macular degeneration (AMD) [11,15,16] and retinitis pigmentosa (RP) [10]. Understanding the role of BMI1 may provide insights into the molecular mechanisms underlying retinal diseases and highlight its potential as a therapeutic target. By understanding native BMI1 levels in the eye, we can start to understand how ocular BMI1 may contribute to various age-related degenerative conditions. Targeting BMI1 may offer new avenues for enhancing retinal regenerative capacity, enhancing mitochondrial function, protecting against oxidative stress, and ultimately contributing to the preservation and restoration of vision in affected individuals.

### Objectives

The objectives of the study were to: (1) determine if the epigenetic regulator BMI1, which plays a critical role in early retinal development, is expressed in the adult eye; (2) describe the ocular biodistribution of BMI1 in the human eye; (3) confirm an age-related decrease in BMI1 levels in retinal tissues; and (4) determine the normal range of BMI1 in adult retinal tissues. This study provides quantitative data on the levels of BMI1 in all ocular tissues and provides a foundation for future studies that may evaluate the dysregulation or change in BMI1 levels in various oxidative stress-associated retinal diseases.

## 2. Materials and Methods

### 2.1. Study Subjects

The study protocol adhered to Association for Research in Vision and Ophthalmology (ARVO) guidance for human tissue and was performed in accordance with an approved IRB protocol. Eye banks were notified of our inclusion and exclusion criteria, and human donor eyes were prospectively collected over one year for tissue analyses for this study alone. Fresh frozen and paraffin-embedded sections of normal donor human eyes (donor ages 20–90 years) were obtained from the NDRI (*n* = 8) and Lions Eye Bank (*n* = 37 eyes, with one eye included per donor) within 48 h of death and frozen after enucleation. We defined young patients as those under 60 years of age and old as those over 60 years of age. The eye globe was immediately placed on ice after transfer from the −80 °C freezer. To dissect the retina and RPE, the anterior segment was first removed by incising the sclera behind the limbus to remove the cornea, iris, lens, and vitreous body. The retina and RPE were then peeled off the eyecup.

#### Selection Process

We included donors aged 20 to 100 with a death to recovery time of 8 h or less. Donors with a current diagnosis of any eye disease, cancer, or sepsis were excluded. 

### 2.2. Eye Dissection and Tissue Lysis

During dissection, the anterior chamber was removed, and the cornea, iris/ciliary body, and lens were collected. The vitreous and retina are carefully peeled off from the underlying RPE/choroid layers, maintaining a clean dissection technique. Peripheral retinal and RPE tissue refer to the extramacular tissue. Tissue lysates were generated via homogenization in ice-cold T-PERTM Tissue protein extraction reagent supplemented with a HaltTM proteinase inhibitor cocktail (ThermoFisher, Waltham, MA, USA). After homogenization, the samples were centrifuged at >10,000× *g* and 4 °C for 10 min and the supernatant was carefully collected. Total protein concentration was measured using a BSA protein assay (ThermoFisher, Waltham, MA, USA). 

### 2.3. Meso Scale Discovery (MSD) Assay

The retinal and RPE lysates were generated and added to a pre-coated plate with a BMI1 capture antibody. The plate was read using an MSD QuickPlex SQ 120 (Meso Scale Discoveries, Rockville, MD, USA), and the data were processed using MSD Workbench 4.0 software (Meso Scale Discoveries, Rockville, MD, USA).

### 2.4. Immunohistochemistry

The eyes were fixed in 4% (vol/vol) paraformaldehyde and then cryopreserved in OCT medium. The fixed eye was sectioned to a thickness of 8 μM. Immunohistochemistry was performed using a rabbit anti-BMI1 antibody (1:800) (Bethyl Lab, Montgomery, TX, USA). An anti-rabbit IgG-HRP conjugate (Vector Labs, Newark, CA, USA) was used as the secondary antibody. A VectorStain Elite ABC kit (Vector Labs, Newark, CA, USA) was used for BMI1 detection. Immunolabeling was captured using an EVOSTM M7000 imaging system (ThermoFisher, Waltham, MA, USA).

### 2.5. Immunofluorescence Assay

To further demonstrate BMI1 expression and localization, immunofluorescence staining was performed using a rabbit anti-BMI1 antibody (1:800) (Bethyl Lab, Montgomery, TX, USA). The secondary antibody was conjugated with AlexaFluor 549 (Invitrogen, Carlsbad, CA, USA). Hoechst 33,342 (Invitrogen, Carlsbad, CA, USA) was used for nuclear staining. Images were visualized and collected using an EVOSTM M7000 imaging system (ThermoFisher, Waltham, MA, USA).

### 2.6. Data Analysis

All data are presented as the mean ± SEM. The student’s *t*-test was performed in GraphPad Prism 10.3 (GraphPad software, Boston, MA USA) and was used to evaluate the differences in BMI1 expression, and a *p*-value < 0.05 was considered statistically significant (*). “ns” represents statistical non-significance.

## 3. Results

### 3.1. Distribution of BMI1 Expression Based on Sex

The mean male and female retinal BMI1 levels were 643.8 ± 127.4 and 741.2 ± 158.5 ng/mg, respectively, without a statistically significant difference (*p* = 0.324). All presented results demonstrate the pooled data of men and women.

### 3.2. BMI1 Expression in the Human Eye

Quantitative levels of BMI1 were highest in the lens and retina and lowest in the cornea, iris, and RPE (Figure 1A). BMI1 protein was found in both the retina (1483.6 ± 191.7 pg/mL) and RPE (296.4 ± 78.1 pg/mL). The BMI1 level was significantly higher in the retina than that in the RPE (*p* < 0.05) (Figure 1B). BMI1 levels in the macula were significantly reduced in the RPE (*p* < 0.05) in older individuals (Figure 1C).

### 3.3. BMI1 Co-Localization in the Retina

BMI1 expression was noted ubiquitously in the GCL (ganglion cell layer), the INL (inner nuclear layer), the ONL (outer nuclear layer), and the RPE (retinal pigment epithelium) by immunofluorescence, with higher levels in the inner than in the outer retinal layers and the RPE (Figure 2). These data indicate that BMI1 is robustly expressed in multiple retinal cell types.

### 3.4. BMI1 Levels in Young and Aged Retinas

To determine BMI1 levels in the human retina of individuals of different ages, we compared young and aged retinal tissues. Our data show that BMI1 is highly expressed in young retinas and is decreased in aged retinas (Figure 3A,B). BMI1 levels in the human retina decrease with age, possibly reflecting changes in retinal cell maintenance, regeneration, and disease susceptibility.

## 4. Discussion

In our study, we found no significant difference in BMI1 levels between men and women. Notably, the lens and retina exhibited the highest levels of BMI1 within the eye, while the retinal pigment epithelium (RPE) displayed considerably lower BMI1 levels compared to the retina. Additionally, we observed an age-related decrease in BMI1 levels in the RPE, specifically in the macula. Furthermore, our results indicate that BMI1 is most abundantly expressed in the inner retina, highlighting its potential importance in retinal health and cellular maintenance. The age-related reduction of RPE BMI1 in the macula may contribute to oxidative stress, cellular dysfunction, and increased risk of AMD. 

Current research suggests that BMI1 is a highly expressed and significant epigenetic regulator in the RPE and neuroretinal macula. BMI1 increases the quantity of antioxidant enzymes, enhances mitochondrial function, increases the production of anti-apoptosis proteins, and promotes cellular renewal. As BMI1 is downregulated in the aging macula, antioxidant activity decreases, which eventually leads to photoreceptor cell death, causing vision loss [8,9]. The role of BMI1 in the human retina is multifaceted and crucial for maintaining retinal health and preventing disease. We focus on the impact of BMI1 function in various retinal diseases, focusing on its regulatory mechanisms, potential as a therapeutic target, and the challenges and opportunities in translating research into clinical practice. Ectopic BMI1 enhances all aspects of mitochondrial function [17] and has no risk of off-target effects in neuronal cells based on RNA-seq [18]. BMI1 helps regulate retinal progenitor cells (RPCs) and retinal stem cells. It promotes the self-renewal and proliferation of these cells and inhibits premature differentiation. This balance is essential for the continuous supply of retinal cells required for repair and maintenance. In the absence of adequate BMI1 function, the renewal and regenerative capacity of the retina is compromised, which can lead to progressive degeneration and loss of vision. 

Loss of BMI1 has been proven to activate oxidative stress and cell death pathways. As BMI1 levels decline with age, antioxidant capacity is reduced, resulting in sequelae of oxidative stress. Heterozygous *Bmi1* knockout mice develop a premature aging phenotype in all organs and ocular findings similar to dry AMD characterized by increased p53, reduced BCL-2, increased apoptosis, lipid peroxidation, and accumulation of neuronal lipofuscin [15]. These antioxidants prevent oxidative stress and complement activation in dry AMD. *Bmi1* knockout in human post-mitotic neurons resulted in amyloidosis, p-tau accumulation, and ultimately, neurodegeneration, all of which are seen in dry AMD in humans [11,16]. *Bmi1* deficiency amplifies p53 activity, accelerating the aging process by promoting the accumulation of oxidative stress, a driving factor for dry AMD [5]. As *Bmi1* expression gradually diminishes in the retina of aging mice and human CNS tissues [8], p53 is upregulated [19]. Ren et al. reported basal BMI1 expression in a normal human pediatric retina, indicating BMI1 involvement in retinal development [20].

Additionally, BMI1’s role in protecting retinal cells from oxidative stress cannot be overstated. The retina, and macula in particular, is highly susceptible to oxidative damage due to its exposure to light and high metabolic rate. BMI1 contributes to the activation of antioxidant pathways, mitigating the harmful effects of ROS (reactive oxygen species). *Bmi1* knockout mice have significantly lower levels of the antioxidants *Sod1*, *Sod2*, and glutathione peroxidase (*Gpx*)*1* and *Gpx3* than their wild-type littermates [21]. Loss of *Sod2* in mouse models generates an AMD phenotype [22] characterized by susceptibility to oxidative stress [23,24], and slow progression of AMD. BMI1 increases SOD2 abundance, which quenches ROS and consequently decreases oxidative damage from multiple activated complement factors [25]. Aged RPEs have increased levels of p53 [19], which downregulates antioxidant genes and augments oxidative damage [5,26], triggering cell death [19,27]. This protective mechanism is vital for preventing oxidative stress-induced retinal diseases such as AMD. *Bmi1* knockout mice have impaired mitochondrial function and increased oxidative stress [21], confirming that BMI1 has a role in maintaining mitochondrial function and redox homeostasis [28]. Ectopic *Bmi1* expression improves all measurements of mitochondrial function [17]. In humans and animal models, mitochondrial dysfunction in photoreceptors has been linked to vision loss.

Our results show that BMI1 is expressed in human retinal tissues. BMI1 is highly expressed in young retinas but is decreased in aged retinas. BMI1 levels in the human retina influence retinal cell function and susceptibility to oxidative stress-induced cell death [7]. BMI1 presents a promising therapeutic target for retinal diseases due to its dual role in cell renewal and protection against oxidative stress. Modulating BMI1 activity could enhance retinal function and encourage retinal cell renewal, offering new treatment avenues for degenerative conditions. For example, gene therapy approaches to upregulate BMI1 in retinal cells may improve their survival and regenerative capacity in AMD. However, targeting BMI1 therapeutically poses several challenges. Reduced BMI1 activity with age may be a cause of reduced retinal function and susceptibility to oxidative stress-induced retinal disease with age. Restoration of BMI1 levels to that of young adulthood may be able to restore cellular renewal pathways and decrease the damage of chronic oxidative stress with targeted delivery. It has been demonstrated that BMI1 overexpression in normal neuronal cells does not induce proliferative or angiogenic effects based on RNA-seq studies and detailed pathway analyses [18]. Moreover, exploring the interaction of BMI1 with other signaling pathways involved in retinal health and disease can provide a holistic understanding of its role. Collaborative efforts between researchers, clinicians, and biotechnologists will be essential in translating these findings into clinical applications.

The age-related decline in retinal function is a multifaceted issue influenced by various biological mechanisms, including the role of BMI1 in cellular maintenance and retinal health. Reduced BMI1 levels may contribute to retinal aging and associated pathologies by increasing cellular senescence, compromising retinal stem cell function, altering the extracellular matrix, regulating oxidative stress response, and increasing the risk of apoptosis. BMI1 is known to inhibit the expression of genes that promote senescence, such as *p16INK4a* and *p21CIP1* [29]. When BMI1 is downregulated, the accumulation of senescent cells in the retina can occur, leading to impaired cellular function and a decline in the regenerative capacity of retinal tissues. BMI1 is critical for maintaining retinal stem cell populations [18]. Reduced expression can compromise the ability of these stem cells to differentiate into retinal neurons and support cells, contributing to the loss of retinal structure and function. This decline in stem cell functionality may be linked to age-related retinal degenerative conditions, such as age-related macular degeneration (AMD) [30]. BMI1 influences extracellular matrix [18] components and signaling pathways that are essential for retinal cell health and survival. Reduced BMI1 may disrupt these pathways, leading to impaired cellular communication and maintenance of retinal architecture, potentially resulting in conditions like retinal detachment or degeneration. BMI1 has a role in modulating the cellular response to oxidative stress [31], which significantly increases with age. Reduced BMI1 levels may impair the ability of retinal cells to cope with oxidative stress, leading to cellular damage and apoptosis. This oxidative damage is a well-known contributor to various retinal pathologies, including diabetic retinopathy and AMD [32]. The decline in BMI1 levels is linked to various retinal pathologies. Conditions such as AMD, retinal vein occlusion, and diabetic retinopathy may be exacerbated by the loss of cellular maintenance and regenerative capacity due to reduced BMI1 expression. The accumulation of senescent cells and impaired response to stressors in the aging retina can further elevate the risk of these pathologies. 

## 5. Conclusions

BMI1 is a key player in maintaining retinal health and protecting against retinal diseases. Its regulatory roles in cell proliferation, differentiation, and oxidative stress response make it a promising therapeutic target. While challenges remain in the therapeutic application of BMI1 modulation, ongoing research and technological advancements offer hope for innovative treatments that could preserve and restore vision in patients with retinal diseases. Few studies have reported on the role of BMI1 in human eye diseases. This study is the first, to our knowledge, to report the expression of BMI1 in the adult human ocular and retinal tissues. BMI1 is downregulated with age in the macula and likely plays a critical role in the self-renewal of adult ocular cells, affecting the retina’s regenerative capabilities and increasing vulnerability to retinal diseases. Loss of BMI1 with age may contribute to mitochondrial dysfunction, increased oxidative stress, and the susceptibility to apoptosis of aged retinal cells. BMI1 levels could serve as a biomarker to assess retinal aging and the risk of developing AMD. A greater understanding of the role of BMI1 in the retina is necessary for developing therapeutic strategies to maintain retinal health and treat age-related diseases with mitochondrial dysfunction such as AMD. 

In summary, the reduction in BMI1 levels in the aging retina can significantly contribute to cellular dysfunction through mechanisms related to senescence, stem cell maintenance, oxidative stress response, inflammation, and matrix remodeling. Understanding these pathways may provide insights into potential therapeutic targets for preserving retinal health and combating age-related retinal diseases. Further research is needed to explore how enhancing BMI1 expression or mimicking its functions may mitigate age-related declines in retinal health.

## Figures and Tables

**Figure 1 cells-13-01672-f001:**
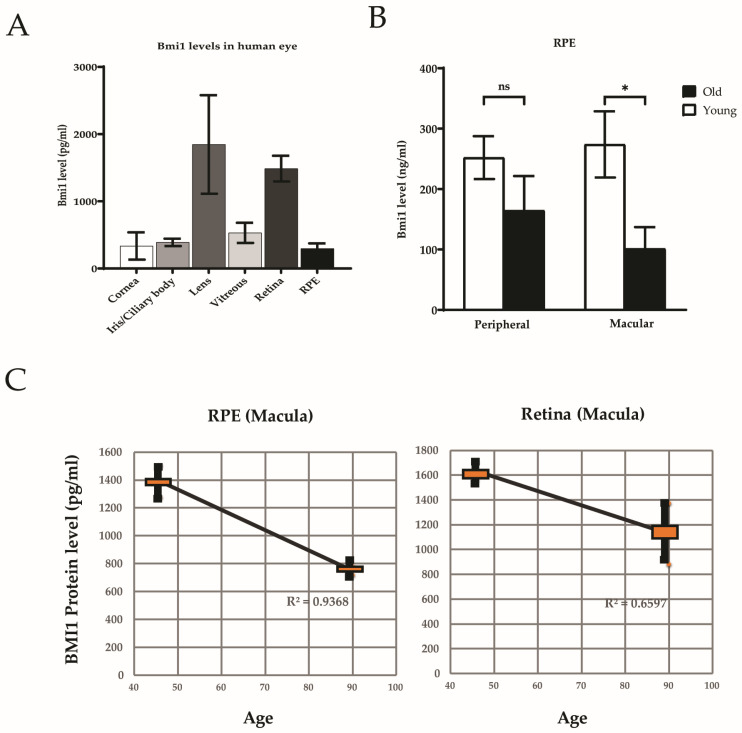
BMI1 protein expression in the human eyes. (**A**) BMI1 levels in all ocular tissues. (**B**) BMI1 levels in the peripheral RPE vs. macula (*n* = 15). Data represent the mean ± SEM (* *p* < 0.05). (**C**) MSD analyses of macular BMI1 protein levels demonstrating an inverse correlation with BMI1 protein levels.

**Figure 2 cells-13-01672-f002:**
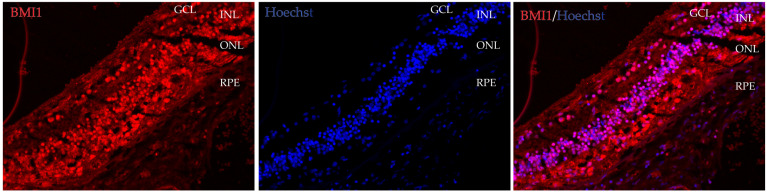
BMI1 localization in the retina in an 88-year-old woman. BMI1 expression was noted ubiquitously in the GCL, INL, ONL, and RPE through immunofluorescence with higher levels in the inner than in the outer retinal layers than the RPE.

**Figure 3 cells-13-01672-f003:**
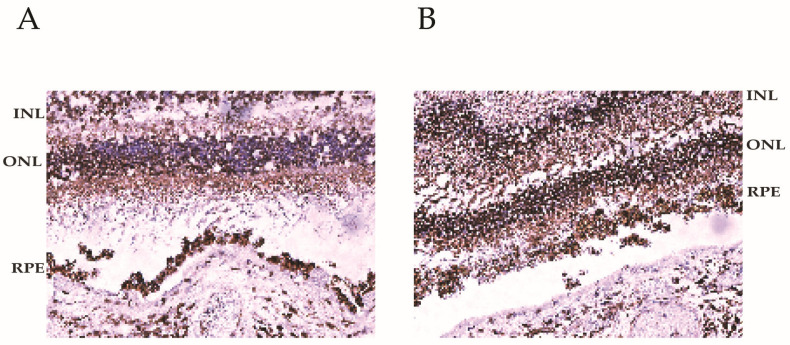
BMI1 expression in the young and aged human retinas. (**A**,**B**). BMI1 was detected via immunohistochemical staining. Loss of pan-retinal BMI1 staining is seen with increased age ((**A**): 76-year-old male, (**B**): 53-year-old male).

## Data Availability

Data are contained within the article.

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
