# Peer review of "The Endogenous Expression of BMI1 in Adult Human Eyes"

_cells, 2024, doi:10.3390/cells13191672_

Round 1

Reviewer 1 Report

Comments and Suggestions for Authors

In this brief manuscript, Lu et al. present an expression study of polycomb protein BMI1 in human eyes using ELISA and immunostaining. The topic is interesting but unfortunately the manuscript does not make a clear or convincing presentation and is therefore very difficult to interpret. Both major and minor issues exist:

Major concerns:

1.     l. 79, details are needed on the source/origin and IRB approval for the AMD eye. This eye is not indicated in the results.

2.     l. 93, the sex distribution of BMI1 levels is not shown in the figure – please add

3.     l. 97, the SEM listed on this line does not match the abstract or Fig. 1B

4.     Methods are missing for dissection of ocular tissues other than retina/RPE. Did the RPE dissection exclude choroidal tissue?

5.     The authors should cite and contrast their findings with PMID: 23559850, which shows low expression of BMI1 in a human pediatric retina

6.     Fig. 1C, how were peripheral, macular, old, and young defined?

7.     Fig. 2 is very hard to interpret – small and low resolution. Please provide the age and location of this individual, and include example images from other eyes. Quantification of BMI1 signal is also needed to enable comparisons between layers. No methods are provided for this immunofluorescence experiment.

8.     Fig. 4C does not make sense – no distribution of ages along the x-axis is shown. Plus, the findings here conflict with the data in Fig. 1C.

Minor issues:

9.     ll. 46-47, reword as no “functions” of BMI1 are assessed here

10.  l. 92, rephrase as this is a section heading

11.  The figure numbers in the text do not match the figures themselves

12.  The discussion proposes BMI1 overexpression as a therapy but oncogenesis is a significant risk; this is not mentioned

13.  The discussion is a nice review of BMI1 function, but says little about what has been learned from this study. Why might BMI1 be expressed in the cell types where it is seen? Why the central/peripheral differences?

Comments on the Quality of English Language

The written English in the manuscript is quite clear.

Author Response

Thank you for your thoughtful review and commentary. We agree with your comments, and they will substantially improve the manuscript. We have made all corrections requested and marked the comments in red.

Major concerns:

  1. l. 79, details are needed on the source/origin and IRB approval for the AMD eye. This eye is not indicated in the results.

Answer: All donor human eyes were obtained from NDRI and Lions Eye Bank (mentioned in lines 71-72). We have IRB approval, and this is now cited (Lines 69-70

  1. l. 93, the sex distribution of BMI1 levels is not shown in the figure – please add

Answer: Thank you, we removed reference to sex distribution as it was not statistically significant and mentioned this in lines 118-21. We removed it from the figure.

  1. l. 97, the SEM listed on this line does not match the abstract or Fig. 1B

Answer: BMI1 levels in the Abstract and Fig.1B were Mean ± SD. We changed the numbers to be SEM in both areas to be consistent and replaced the numbers in Fig. 1B.

  1. Methods are missing for dissection of ocular tissues other than retina/RPE. Did the RPE dissection exclude choroidal tissue?

Answer: Thank you for bringing this to our attention. We have added ocular tissue dissection in Methods 2.2 Eye dissection and Tissue lysis (lines 82-86). The RPE values includes choroidal tissue, and this was clarified in this section.

  1. The authors should cite and contrast their findings with PMID: 23559850, which shows low expression of BMI1 in a human pediatric retina

Answer: Thank you for your suggestion. We have added this reference in the manuscript (lines 198-99). The referenced paper has a qualitative comparison of normal versus abnormal retinal tissue based on histopathology. BMI1 expression in the retina and eye is extremely low compared to other tissues. (https://www.proteinatlas.org/ENSG00000168283-BMI1/tissue ).

  1. Fig. 1C, how were peripheral, macular, old, and young defined?

Answer: The peripheral retina and RPE refers to the areas of the retina found outside of the macula. In our database, “Young” refers to those under 60 years old, and “old” refers to those over 60 years old. This is now specified in the manuscript in lines 73-74 and 132-133.

  1. Fig. 2 is very hard to interpret – small and low resolution. Please provide the age and location of this individual, and include example images from other eyes. Quantification of BMI1 signal is also needed to enable comparisons between layers. No methods are provided for this immunofluorescence experiment.

Answer: The image taken from an 88-year-old woman close to macula, and the manuscript was updated to reflect this (line 152). We wanted to present BMI1 expression in the entire retina, so a low magnification image was selected. We can best quantify expression with immunohistochemistry but are unable to quantify the BMI1 signal in human retina due to the small numbers of human samples and poor fixation of other slides of humans which prevented staining of many of the samples we collected. Per your suggestion, we have added immunofluorescence assay methodology to Methods section as section 2.5 (Lines 106-111).

  1. Fig. 4C does not make sense – no distribution of ages along the x-axis is shown. Plus, the findings here conflict with the data in Fig. 1C.

Answer:

Thank you for pointing out that the Figure was not clear. We present 3 figures in the manuscript.  In Figure 3C, we have the ages along the x axis, but the image was small. To address this, we have enlarged the Figure.  Additionally, we included a graph of individuals with and without eye disease, which was a different dataset that Figure 3C (all normal).  To avoid confusion, we excluded this graph and kept the discussion to just the normal human eyes.

Minor issues:

  1. ll. 46-47, reword as no “functions” of BMI1 are assessed here

Answer: Thank you for this comment. We have removed any reference to functions and have replaced this sentence with one describing the natural levels of BMI1 in the human retina (lines 50-52).

  1. l. 92, rephrase as this is a section heading

Answer: Changed to 3.1. Age distribution in this study is presented in Figure 1 (line 118).

  1. The figure numbers in the text do not match the figures themselves

Answer: Thank you for pointing this out. We have updated all figure references for consistency.

  1. The discussion proposes BMI1 overexpression as a therapy but oncogenesis is a significant risk; this is not mentioned

Answer: Luckily, this is not a risk in normal neuronal cells, but only in cancer cells.  We have included this statement: BMI1 overexpression in normal neuronal cells does not induce any proliferative or angiogenic effects, based on RNA-seq studies and detailed pathway analyses (Ganapathi et al. (2018) 8:7464 | DOI:10.1038/s41598-018-25921-8). This is in lines 229-231

  1. The discussion is a nice review of BMI1 function, but says little about what has been learned from this study. Why might BMI1 be expressed in the cell types where it is seen? Why the central/peripheral differences?

Answer: Thank you.  We have expanded the section on what has been learned in the study in the 4. Discussion section as follows: “In our study, we found no significant difference in BMI1 levels between men and women. Notably, the lens and retina exhibited the highest levels of BMI1 within the eye, while the retinal pigment epithelium (RPE) displayed considerably lower BMI1 levels compared to the retina. Additionally, we observed an age-related decrease in BMI1 levels in the RPE specifically in the macula. Furthermore, our results indicate that BMI1 is most abundantly expressed in the inner retina, highlighting its potential importance in retinal health and cellular maintenance. The loss of RPE BMI1 with age in the macula may be a reason for age-related oxidative stress and damage and increase the risk for AMD.” This is in lines 163-170.

Reviewer 2 Report

Comments and Suggestions for Authors

The authors effectively summarizes the main findings of the study and highlights the significance of BMI1 expression in the human retina and RPE. It clearly identifies the observed differences in BMI1 levels between inner and outer retinal layers and notes the potential implications for age-related changes. The manuscript is well-written and scientifically sound. It has potentially high interest to readers interested in the area of retina aging research. I would suggest that the authors consider the following points as they revise their manuscript and continue their work in this important research area.

My specific comments are mentioned below:

1. The objectives of this study are absent. After the introduction, one paragraph for the objectives statement is necessary.

2. Introduction needs minor revision. Author could check the recently published studies.

3. Specifying the exact type of study (e.g., observational, experimental) might be helpful.

4. Consider adding details on any inclusion/exclusion criteria for the donor eyes (if any) to clarify the selection process.

5. The data demonstrating significantly higher BMI1 levels in the young macula compared to the aged macula provide valuable insights into the impact of aging on BMI1 expression. This trend is suggestive of BMI1’s role in age-related retinal changes. The manuscript should expand on potential mechanisms underlying this age-related decline. Specifically, how might reduced BMI1 levels contribute to retinal aging, and could this be linked to specific retinal pathologies or reduced cellular maintenance?

6. The manuscript correctly reports a non-significant difference in BMI1 levels between sexes (p=0.324), which supports focusing on age-related variations. For clarity and reproducibility, it would be beneficial to provide more detail on the statistical methods employed. Specifically, a description of the statistical tests used for comparing BMI1 levels and assessing co-localization would strengthen the robustness of the results.

7. The discussions are poorly referenced with literature that does not support the statements made in the text.

8. In the conclusions paragraph effectively highlights the key findings regarding BMI1's role in retinal health and disease. However, consider breaking down complex sentences to improve readability and clarity. For example, the sentence starting with "While little is known about the role of BMI1 in human retinal disease..." could be rephrased for better flow.

Author Response

Thank you for pointing this out. We agree with this comment and have made following the changes and are marked in red.

  1. The objectives of this study are absent. After the introduction, one paragraph for the objectives statement is necessary.

Answer: Thank you for identifying this omission. We have added a section describing the study objectives as follows:

Objectives

The objectives of the study are to: 1) Determine if the epigenetic regulator BMI1, which plays a critical role in early retinal development, is expressed in the adult eye; 2) Describe the ocular biodistribution of BMI1 in the human eye; 3) Confirm an age-related decrease in BMI1 levels in ocular tissues; 4) Determine the normal range of BMI1 in adult retinal tissues. This study provides quantitative data on the levels of BMI1 in all ocular tissues and provides a foundation for future studies that may evaluate the dysregulation or change in BMI1 levels in various oxidative stress-associated retinal diseases.

This is in lines 56-65.

  1. Introduction needs minor revision. Author could check the recently published studies.

Answer: We have included comments from the more recent literature and discussed the role of BMI1 is neural stem cell renewal and cellular homeostasis. We have added a reference from the cardiac literature:

https://www.thelancet.com/journals/ebiom/article/PIIS2352-3964(20)30569-7/fulltext

 We also included the findings of BMI1 knockout mice (premature cataracts and retinal degeneration) and have included these additional references in the manuscript (line 42-44)

  1. Specifying the exact type of study (e.g., observational, experimental) might be helpful.

Answer: This is a prospective experimental study.

  1. Consider adding details on any inclusion/exclusion criteria for the donor eyes (if any) to clarify the selection process.

Answer:  We included donors ages 20 to 100 with death to recovery time 8 hours or less. We excluded subjects with a current diagnosis of cancer or sepsis.  This is now included as a selection process section in lines 79-81.

  1. The data demonstrating significantly higher BMI1 levels in the young macula compared to the aged macula provide valuable insights into the impact of aging on BMI1 expression. This trend is suggestive of BMI1’s role in age-related retinal changes. The manuscript should expand on potential mechanisms underlying this age-related decline. Specifically, how might reduced BMI1 levels contribute to retinal aging, and could this be linked to specific retinal pathologies or reduced cellular maintenance?

Answer: The age-related decline in retinal function is a multifaceted issue influenced by various biological mechanisms, including the role of BMI1 in cellular maintenance and retinal health. Reduced BMI1 levels may contribute to retinal aging and associated pathologies by increasing cellular senescence, compromising retinal stem cell function, altering the extracellular matrix, regulating oxidative stress response, and risk for apoptosis. BMI1 is known to inhibit the expression of genes that promote senescence, such as p16INK4a and p21CIP1. When BMI1 is downregulated, the accumulation of senescent cells in the retina can occur, leading to impaired cellular function and a decline in the regenerative capacity of retinal tissues. BMI1 is critical for maintaining retinal stem cell populations. Reduced expression can compromise the ability of these stem cells to differentiate into retinal neurons and support cells, contributing to the loss of retinal structure and function. This decline in stem cell functionality may be linked to age-related retinal degenerative conditions, such as age-related macular degeneration (AMD). BMI1 influences extracellular matrix components and signaling pathways that are essential for retinal cell health and survival. Reduced BMI1 may disrupt these pathways, leading to impaired cellular communication and maintenance of retinal architecture, potentially resulting in conditions like retinal detachment or degeneration. BMI1 has a role in modulating the cellular response to oxidative stress, which significantly increases with age. Reduced BMI1 levels may impair the ability of retinal cells to cope with oxidative stress, leading to cellular damage and apoptosis. This oxidative damage is a well-known contributor to various retinal pathologies, including diabetic retinopathy and AMD. The decline in BMI1 levels is linked to various retinal pathologies. Conditions such as AMD, retinal vein occlusion, and diabetic retinopathy may be exacerbated by the loss of cellular maintenance and regenerative capacity due to reduced BMI1 expression. The accumulation of senescent cells and impaired response to stressors in the aging retina can further elevate the risk of these pathologies.

  1. The manuscript correctly reports a non-significant difference in BMI1 levels between sexes (p=0.324), which supports focusing on age-related variations. For clarity and reproducibility, it would be beneficial to provide more detail on the statistical methods employed. Specifically, a description of the statistical tests used for comparing BMI1 levels and assessing co-localization would strengthen the robustness of the results.

Answer: Thank you for the clarification. We have added statistical methods in the Results section as follows: “The Student’s t-test was performed in GraphPad prism and was used to evaluate the difference in BMI1 expression, and a p < 0.05 was considered statistically significant (*). p < 0.01 (**) or p < 0.001 (***) was considered very significant. “ns” represents statistical non-significance.” In lines 113-116

  1. The discussions are poorly referenced with literature that does not support the statements made in the text.

Answer: We have added multiple references in the Discussion section and corrected incorrect references.

  1. In the conclusions paragraph effectively highlights the key findings regarding BMI1's role in retinal health and disease. However, consider breaking down complex sentences to improve readability and clarity. For example, the sentence starting with "While little is known about the role of BMI1 in human retinal disease..." could be rephrased for better flow.

Answer: Thank you for the comment, we have revised the writing throughout the document to improve readability and flow.

Round 2

Reviewer 1 Report

Comments and Suggestions for Authors

In this revision, the authors have made efforts to address reviewer comments. However, some issues remain and new ones have arisen with the changes made:

1.     l. 70, I would not consider this a prospective study – subjects were not recruited specifically for this study.

2.     l. 118, the heading text about age is not relevant for this section about sex

3.     Fig. 1, are the data in Fig. 1B simply repeating from Fig. 1A? This is not clear. If so, Fig. 1B should be deleted.

4.     l. 148 mentions that panel 1C shows peripheral retina data but the figure itself and text states RPE. This needs to be clarified. Also, units are listed as pg/mg for this figure but no others. Please explain. Please also indicate the number of individuals used in this analysis (is it the full n=45 from the Methods?)

5.     Sections 3.2 and 3.3 describe the same data and should be combined.

6.     Fig. 2, Hoechst is misspelled (and is not mentioned in the Methods)

7.     Fig. 3C, I still do not understand this figure – a scatterplot of ages vs. BMI1 levels would be much more informative. If this is ELISA data, it should logically be combined as a panel within Fig. 1.

8.     The new final paragraph of the discussion only cites one reference; much more citation of the statements is needed.

Author Response

Point-by-point Revisions to Reviewer 1 Comments (round2).

Thank you for your thoughtful review and commentary. We agree with your comments, and they will substantially improve the manuscript. We have made all corrections requested and marked the comments in red.

  1. 70, I would not consider this a prospective study – subjects were not recruited specifically for this study.

Answer:  Eye banks were notified of our inclusion and exclusion criteria, and human donor eye were prospectively collected over one year for tissue analyses for this study alone.  We will describe this method of collection in the manuscript in lines 70-72.

  1. 118, the heading text about age is not relevant for this section about sex

Answer: Thank you for pointing this out. We change the heading to sex (line 119)

  1. 1, are the data in Fig. 1B simply repeating from Fig. 1A? This is not clear. If so, Fig. 1B should be deleted.

Answer: Yes.  Figure 1B was deleted, and a new Figure 1 was included before Line 141.

  1. 148 mentions that panel 1C shows peripheral retina data but the figure itself and text states RPE. This needs to be clarified. Also, units are listed as pg/mg for this figure but no others. Please explain. Please also indicate the number of individuals used in this analysis (is it the full n=45 from the Methods?)

Answer: Thank you for pointing this out. Fig 1b is RPE, and we corrected the figure legend accordingly and included n=15 in this analysis in line 143

  1. Sections 3.2 and 3.3 describe the same data and should be combined.

Answer: Thank you for the suggestion. We combined section 3.2 and 3.3 together (line 123-129).

  1. 2, Hoechst is misspelled (and is not mentioned in the Methods)

Answer: Thank you for pointing this out (Fig 2). We corrected spelling and added to the Methods (Line111),

  1. 3C, I still do not understand this figure – a scatterplot of ages vs. BMI1 levels would be much more informative. If this is ELISA data, it should logically be combined as a panel within Fig. 1.

Answer: Thank you.  We will replace this as a scatterplot of age vs. BMI1 with the correlation line and add it to Figure 1 as Figure 1C (line 141).

  1. The new final paragraph of the discussion only cites one reference; much more citation of the statements is needed.

Answer: Five additional references have been added to this paragraph (line 234,240,241,245,248).